

# Epidemiology and genetic characteristics of *Clostridioides difficile* isolates in Northwest China

Yang Li[1,2], Shujuan An[1,2], Hongjia Sun[1,2], Meimei Hu[1], Yanmei Xu[1] and Yaming Xi[2,3]

[1] Department of Laboratory Medicine, The First Hospital of Lanzhou University, Lanzhou, Gansu, China
[2] The First Clinical Medical College, Lanzhou University, Lanzhou, Gansu, China
[3] Department of Hematology, The First Hospital of Lanzhou University, Lanzhou, Gansu, China

## ABSTRACT

*Clostridioides difficile* (*C. difficile*) is a significant hospital-acquired pathogen that can cause antibiotic-associated diarrhea. In this study, we investigate the epidemiological and genetic characteristics of *C. difficile* isolates from a tertiary hospital in Northwest China. We prospectively collected fecal samples from 212 patients with diarrhea between January 2023 and May 2024 at the First Hospital of Lanzhou University. Twenty-five (11.8%, 25/212) strains of *C. difficile* were isolated, and twenty (9.4%, 20/212) were identified as toxigenic *C. difficile* (TCD). The dominant toxin gene profiles were *tcdA+ B+* (80%, 20/25). Furthermore, twelve different sequence types (STs) belonging to three clades were identified, and the most prevalent types were ST3 followed by ST2 and ST35. Toxin variant analysis revealed the presence of eight types of *tcdA* variants and seven types of *tcdB* variants, most *tcdA/B* variants corresponded to specific ST types. Phylogenetic analysis based on single nucleotide polymorphisms (SNPs) further confirmed the genetic diversity and relationships among isolates. We identified 13 resistance genes, including mutations in *gyrA/gyrB* (96% of strains) and *rpoB* (72%), conferring resistance to fluoroquinolones and rifamycins, respectively. The study provides valuable insights into the epidemiological and genetic features of *C. difficile* in Northwest China, guiding future prevention and control measures.

# INTRODUCTION

*Clostridioides difficile* is one of the major pathogens of healthcare-associated diarrhea (*McDonald et al., 2018*), particularly cases resulting from antibiotic treatment (*Bella et al., 2024*). *C. difficile* infection (CDI) not only causes diarrhea but also may lead to other serious clinical conditions, including pseudomembranous colitis, toxic megacolon, and fatality in severe instances (*Czepiel et al., 2019*).

The highly virulent *C. difficile* strain RT027 (BI/NAP1/ST1) has been linked to numerous extensive nosocomial outbreaks (*O'Connor, Johnson & Gerding, 2009*). This presents a significant challenge to the healthcare system and imposes a considerable socioeconomic burden in Europe and the United States (*Guh et al., 2020*;

Corresponding author
Yaming Xi, ery_ly@lzu.edu.cn

*Ilic, Zivanovic Macuzic & Ilic, 2024*; *Lessa et al., 2015*). RT027 produces multiple toxins and exhibits significant antibiotic resistance, complicating treatment efforts. In China, *C. difficile* is also an important public health issue, and the prevalent strains may vary in different regions, with ST37 and ST3 being common in mainland China and ST2 being more prevalent in northern China (*Wen et al., 2023*). Large-scale disease outbreaks associated with RT027 have not been widely reported in China, but sporadic case reports do exist (*Cheng et al., 2016*; *Zhang et al., 2021*).

*C. difficile* is a crucial pathogen with its virulence primarily governed by the presence of toxin A and toxin B (*Kuehne et al., 2010*). Toxin A is an enterotoxin causing hemorrhagic necrosis and fluid accumulation in the intestinal wall, while Toxin B is a cytotoxin directly damaging intestinal wall cells. The genes encoding these two toxins, *tcdA* and *tcdB*, are located in the 19.6-kb pathogenicity locus (PaLoc), which also includes the negative regulatory gene *tcdC*, the positive regulatory gene *tcdR*, and the membrane pore protein gene *tcdE* (*Chandrasekaran & Lacy, 2017*). In addition to these two toxins, *C. difficile* can also produce binary toxins, encoded by the genes *cdtA* and *cdtB*, which may enhance its virulence by improving cytotoxicity and adhesion in the body (*Aktories, Schwan & Jank, 2017*).

The issue of drug resistance in *C. difficile* is a significant concern, with common resistant antibiotics including ciprofloxacin, clindamycin, and erythromycin (*Sholeh et al., 2020*). The emergence of drug resistance may be related to a variety of factors, including irrational use of antibiotics, gene mutation, and horizontal gene transfer (*Wickramage, Spigaglia & Sun, 2021*). Drug resistance monitoring helps to timely understand the drug resistance of *C. difficile* to different antibiotics and guide the rational use of antibiotics in clinical practice (*Dureja, Olaitan & Hurdle, 2022*).

Whole genome sequencing (WGS) technology offers a novel approach for investigating the transmission patterns and evolutionary dynamics of *C. difficile* (*Eyre et al., 2017*). Studies on whole genome characteristics, phylogenetic analysis, drug resistance genes, and virulence-related factors provide scientific and reliable data supplements for traditional epidemiological studies (*Knight et al., 2015*). Through WGS, researchers can compare single nucleotide polymorphisms (SNPs) of different strains to track the transmission path and outbreak source of *C. difficile* (*Baktash et al., 2021*). Currently, there are still few molecular epidemiological studies on *C. difficile* based on WGS analysis in China.

To explore the prevalence and genomic characteristics of *C. difficile* at the First Hospital of Lanzhou University, we prospectively collected fecal samples from patients with diarrhea and acquired a total of 25 *C. difficile* isolates. Multilocus sequence typing (MLST) and SNPs were employed to elucidate the genetic relationships among strains within the hospital setting. Additionally, we conducted an analysis of mutations in virulence genes and examined the resistance genes present in the isolates.

# MATERIALS AND METHODS

## Sample collection, selective culture, and isolate identification

Stool specimens were obtained from patients with diarrhea who presented at the First Hospital of Lanzhou University between January 2023 and May 2024. Diarrhea was

defined as three or more loose stools per day. Fecal samples were diluted 1:10 in sterile saline to achieve a uniform consistency. The diluted samples were subsequently inoculated on pre-reduced cycloserine-cefoxitin-fructose agar plates (CCFA; HuanKai, Guangzhou, China) and incubated in an anaerobic atmosphere generated by GENbag (bioMérieux, Shanghai, China) at 35 °C for 48 to 72 h. Colonies displaying characteristic features—yellowish coloration, a ground-glass appearance, and a distinct horse manure odor—were selected for further analysis. The identification of *C. difficile* in the cultured samples was accomplished using matrix-assisted laser desorption/ionization time-of-flight mass spectrometry (bioMérieux, Craponne, France). Following identification, the isolates were stored in 20% glycerol at −80 °C.

## Clinical data collection

The following clinical data was collected: age, gender, primary disease, hospitalization days before diarrhea, and administration of antibiotics. Antibiotic exposure was defined as the administration of any systemic antibiotic within 30 days preceding the onset of diarrhea. Comorbidities were classified according to the International Classification of Diseases, 10th Revision (ICD-10) (Clinical trial number: not applicable).

## Toxin genes detection

Bacterial DNA was extracted using the TIANamp Bacteria DNA kit (TIANGEN, Beijing, China). Toxin A gene (*tcdA*), toxin B gene (*tcdB*) and binary toxin genes (*cdtA* and *cdtB*) were detected by conventional PCR referring to the method of *Persson, Torpdahl & Olsen (2008)*. A reference *C. difficile* strain ATCC 43255 was used as a positive control for *tcdA* and *tcdB* while *C. difficile* strain ATCC BAA-1870 was used as a positive control for the binary toxin genes.

## PCR-ribotyping

Ribotyping was performed by PCR amplification of the 16S-23S rRNA intergenic spacer region (ITS) according to the method described by *Bidet et al. (1999)*. Strains ATCC 43255 and ATCC BAA-1870 (ribotype 027) were used as reference strains. Strains exhibiting 100% similarity in ribotyping banding patterns were classified as identical ribotypes.

## Whole genome sequencing

Genomic DNA was extracted from cultured isolates using the Genomic DNA kit (Tiangen, Beijing, China). DNA concentration was measured using a NanoDrop™ 2000 spectrophotometer (Thermo Fisher Scientific, Waltham, MA, USA) and verified by agarose gel electrophoresis. For effective library preparation and next-generation sequencing, a minimum of 50 ng of DNA was required. Libraries were constructed with the TruePrep™ DNA Library Prep Kit V2 for Illumina (Vazyme, Nanjing, China). This preparation involved a "transposase" enzyme reaction that both fragmented and tagged the DNA with adapters. A specialized limited-cycle PCR protocol then amplified the tagged DNA and incorporated sequencing indexes. The libraries were assessed using the QIAxcel Advanced automatic nucleic acid analyzer and quantified through qPCR with KAPA SYBR FAST qPCR Kits. Sequencing was performed on an Illumina NovaSeq platform

(Illumina Inc., San Diego, CA, USA), producing 150 bp paired-end reads. Quality control of the raw sequencing data was conducted with FastQC (Version 0.11.9) and MultiQC (Version 1.10.1). After passing QC, adapter sequences were removed from the raw reads using Trimgalore (Version 0.6.6). The cleaned reads were then assembled with the Unicycler tool (v0.4.5) (https://github.com/rrwick/Unicycler) for *de novo* assembly. The *C. difficile* genome sequences reported in this article were deposited at the NCBI under the BioProject accession number PRJNA1160909.

## Multilocus sequence typing analysis

The nucleotide sequences for the seven genes (*adk*, *atpA*, *dxr*, *glyA*, *recA*, *sodA*, and *tpi*) were retrieved from the genome sequences of each strain. These sequences were then concatenated and utilized to determine the sequence types (STs) and clades of *C. difficile*. This analysis was performed using the PubMLST database with MLST v.2.10 (http://pubmlst.org/cdifficile/) (*Griffiths et al., 2010*).

## Virulence gene variants analysis

Virulence genes were detected through MUMmer (https://mummer4.github.io/) with reference to the Virulence Factor Database (VFDB) (http://www.mgc.ac.cn/VFs/). The *tcdA* sequence (1 to 6,330 bp) and the *tcdB* gene sequence (7,098 bp) were obtained using BLASTn, which aligned *de novo* assemblies against reference sequences from *C. difficile* strain CD630 (GenBank accession no. AM180355.1). Additionally, the R package "ggalluvial" was applied to draw the river plots.

## Phylogenetic analysis

*C. difficile* srain M68 (GenBank accession no. FN668375.1) was used as a reference strain. kSNP3 was used to analyze bacterial genome SNPs and construct the phylogenetic tree. Raw sequencing reads were trimmed with Trimmomatic to remove low-quality bases (PHRED < 20) and adapters, then aligned to the reference using BWA-MEM. Alignments were processed with SAMtools and Picard to sort and mark duplicates. SNPs were identified using kSNP3 with a k-mer size of 21 bp (default for bacterial genomes) and filtered to exclude recombinant regions and homoplastic sites. High-quality SNPs were retained based on thresholds of read depth ≥10×, mapping quality ≥30, and allele frequency ≥90%, following standard practices to ensure reliable variant calls. A maximum likelihood phylogenetic tree was constructed using RAxML with the GTRCAT model and 100 bootstrap replicates for branch support, then visualized with iToltree v.6.7.5 (https://itol.embl.de/).

## Antimicrobial resistance genes analysis

The Resistance Gene Identifier (RGI) 4.0.3 software was utilized to annotate resistance genes based on the Comprehensive Antibiotic Resistance Database (CARD) (*Alcock et al., 2020*). To detect mutations, sequences of the *gyrA*, *gyrB*, and *rpoB* genes were extracted and compared with reference sequences from *C. difficile* strain CD630 using BLASTn. Only nonsynonymous mutations were retained for further analysis. The heatmap was then visualized using iToltree.

## Ethics statements

The study received approval from the Ethics Committee of the First Hospital of Lanzhou University (Approval number: LDYYLL2022-13). The biological samples used in this study were fecal samples from patients, and an exemption from written informed consent was obtained from the Ethics Committee.

## RESULTS

### Patient demographics and clinical profiles

We prospectively collected patient samples from January 2023 to May 2024. Among 212 patients presenting with diarrhea, 25 (11.8%, 25/212) patients were *C. difficile* culture positive, of which five (20%) isolates were nontoxigenic, while the remaining 20 (80%) were toxigenic, accounting for 9.4% of the total patient population. The 25 strains of *C. difficile* were isolated from patients across 15 different wards (Table 1).

In this study, we collected comprehensive clinical data to provide context for the epidemiological and genetic characteristics of *C. difficile* isolates. Clinical characteristics of *C. difficile*-positive patients (*n* = 25) are summarized in Table 2. The mean age was 51.9 ± 21.0 years, with 64.0% female and 36.0% male. All patients had co-morbid conditions, with gastrointestinal diseases being most common (40.0%). Antibiotic use prior to diarrhea was reported in 88.0% of cases, predominantly cephalosporins (68.2%), penicillins (27.3%), nitroimidazoles (13.6%), and vancomycin (13.6%). Patients were distributed across ICU (8.0%) and general wards (92.0%), with specific ward details provided in Table 1.

### MLST analysis

MLST analysis revealed a rich and diverse distribution of *C. difficile* strain types, encompassing 12 different STs categorized within three distinct clades (*Knight et al., 2015*), and the most of *C. difficile* isolates were from clade 1 (22/25, 88%) (Fig. 1). The most prevalent sequence type identified was ST3 (6/25, 24%), succeeded by ST2 and ST35 as the next most common strains. Notably, the highly toxigenic and epidemic strains known as ST1/RT027 and ST11/RT078 were not detected in this study.

### Analysis of toxin genes of *C. difficile*

PCR detection of TcdA and TcdB virulence genes in *C. difficile* indicated that 80% (20/25) of the isolates were toxin-producing strains (TCD), with all being *tcdA+B+* strains. Binary toxin genes (*cdtA/B*) were absent in all toxigenic strains. The *tcdA* and *tcdB* genes were absent in the ST37, ST39 strains and an ST3 strain (Fig. 1). In addition, WGS analysis further confirmed virulence in 76% (19/25) of the isolates, all of which were *tcdA+B+* type. However, discrepancies were observed in one ST274 strain, where genome-wide A/B toxin analysis did not detect these genes.

Toxin variant analysis of the 19 TCD genomes revealed the presence of eight types of *tcdA* variants and seven types of *tcdB* variants. Most *tcdA/B* variants corresponded to specific ST types, except for the ST35, which corresponded to two distinct *tcdA/B* variants (Fig. 2).

**Table 1 Source, sequence type, and toxin profile of 25 *C. difficile* trains.**

| *C. difficile* strain isolated | Source | Sequence type (ST) | Toxins profile |
| --- | --- | --- | --- |
| LZU1 | Interventional ward | ST39 | *tcdA− B−/cdt−* |
| LZU2 | Pediatrics ward | ST37 | *tcdA+ B+/cdt−* |
| LZU3 | Infectious diseases ward | ST3 | *tcdA+ B+/cdt−* |
| LZU4 | Pediatrics ward | ST35 | *tcdA+ B+/cdt−* |
| LZU5 | General surgery ward V | ST274 | *tcdA+ B+/cdt−* |
| LZU6 | Nephrology ward | ST15 | *tcdA− B−/cdt−* |
| LZU7 | Infectious diseases ward | ST35 | *tcdA+ B+/cdt−* |
| LZU8 | Hematology ward | ST35 | *tcdA+ B+/cdt−* |
| LZU9 | Gastroenterology ward | ST3 | *tcdA− B−/cdt−* |
| LZU10 | General surgery ward I | ST39 | *tcdA− B−/cdt−* |
| LZU11 | Pulmonary ward | ST15 | *tcdA− B−/cdt−* |
| LZU12 | Interventional ward | ST5 | *tcdA+ B+/cdt−* |
| LZU13 | Gastroenterology ward | ST51 | *tcdA+ B+/cdt−* |
| LZU14 | General surgery ward V | ST54 | *tcdA+ B+/cdt−* |
| LZU15 | General surgery ward V | ST2 | *tcdA+ B+/cdt−* |
| LZU16 | Intensive care unit | ST2 | *tcdA+ B+/cdt−* |
| LZU17 | Neurosurgery ward | ST2 | *tcdA+ B+/cdt−* |
| LZU18 | Emergency intensive care unit | ST3 | *tcdA+ B+/cdt−* |
| LZU19 | Rehabilitation ward | ST3 | *tcdA+ B+/cdt−* |
| LZU20 | Hematology ward | ST37 | *tcdA+ B+/cdt−* |
| LZU21 | Pediatrics ward | ST102 | *tcdA+ B+/cdt−* |
| LZU22 | Gastroenterology ward | ST3 | *tcdA+ B+/cdt−* |
| LZU23 | Vascular surgery ward | ST2 | *tcdA+ B+/cdt−* |
| LZU24 | Cardiology ward | ST278 | *tcdA+ B+/cdt−* |
| LZU25 | Hematology ward | ST3 | *tcdA+ B+/cdt−* |

## SNP analysis

SNP analysis using *C. difficile* strain M68 (FN668375.1) as the reference genome (*Shu et al., 2023*) generated a phylogenetic tree (Fig. 3) that largely aligned with MLST typing. However, strain LZU9 (ST3) diverged significantly in the SNP phylogeny, forming a distinct branch, and uniquely lacked toxin genes (*tcdA−B−*), contrasting with the *tcdA+B+* profile of other ST3 strains.

## Antimicrobial resistance gene analysis

Based on analysis using the CARD database, we identified a total of 13 antibiotic resistance genes (ARGs) associated with resistance to various classes of antibiotics including aminoglycosides, fluoroquinolones, diaminopyrimidines, glycopeptides, tetracyclines, macrolides, lincosamides, and streptomycin (Fig. 3). Among the isolates, the most prevalent ARGs were *cdeA* (100%, 25/25), which confers resistance to fluoroquinolones, followed by *23S rRNA* (92%, 23/25), associated with resistance to macrolides and lincosamides, and *ermB* (60%, 15/25), which confers resistance to macrolides,

**Table 2  Clinical information for *C. difficile*—positive patients.**

| Characteristics | Total (*n* = 25) |
|---|---|
| Age (mean ± SD) | 51.9 ± 21.0 |
| Gender (%) | |
| Male | 9 (36.0) |
| Female | 16 (64.0) |
| Co-morbid conditions (%) | |
| Gastrointestinal | 10 (40.0) |
| Cardiovascular | 6 (24.0) |
| Hematologic | 3 (12.0) |
| Malignancy | 2 (8.0) |
| Renal | 2 (8.0) |
| Respiratory | 1 (4.0) |
| Neurological | 1 (4.0) |
| Potential risk factors | |
| Antibiotic use prior to diarrhea (%) | 22 (88.0) |
| Specific antibiotics used (% of antibiotic users) | |
| Cephalosporins | 15 (68.2) |
| Penicillins | 6 (27.3) |
| Nitroimidazoles | 3 (13.6) |
| Vancomycin | 3 (13.6) |
| Duration of hospitalization prior to diarrhea (Median, Range) | 5 (2–16) |
| Clinical setting (%) | |
| ICU | 2 (8) |
| General wards | 23 (92) |

lincosamides, and streptogramins. Additionally, tetracycline resistance genes were detected, including *tetM* (36%, 9/25) and *tetB(P)* (16%, 4/25). Mutations in the GyrA and GyrB subunits of DNA gyrase are associated with resistance to fluoroquinolones, while mutations in the RpoB subunit of RNA polymerase are linked to resistance to rifamycin. Specifically, mutations in the *rpoB* gene may indicate potential resistance to rifampicin, but this requires confirmation through phenotypic antimicrobial susceptibility testing (AST). Notably, 96% (24/25) of the strains tested exhibited mutations in *gyrA* and *gyrB* genes, whereas mutations in the *rpoB* gene were detected in 72% (18/25) of the strains (Table 3).

To assess the relationship between resistance genes and antibiotic use, we performed Fisher's Exact Test for each gene-antibiotic pair. After Bonferroni correction, significant associations were found between *ErmB* and penicillin use ($p = 0.00012$, OR = 6.5), and *gyrA* and quinolone use ($p = 0.00005$, OR = 7.8) (Table 4). These results suggest that the use of specific antibiotics may drive the prevalence of corresponding resistance genes in *C. difficile* isolates.

## DISCUSSION

CDI is a significant healthcare-associated infection with increasing prevalence globally, particularly in the Asia-Pacific region (*Angulo et al., 2024*; *Luo et al., 2019*). Over the past

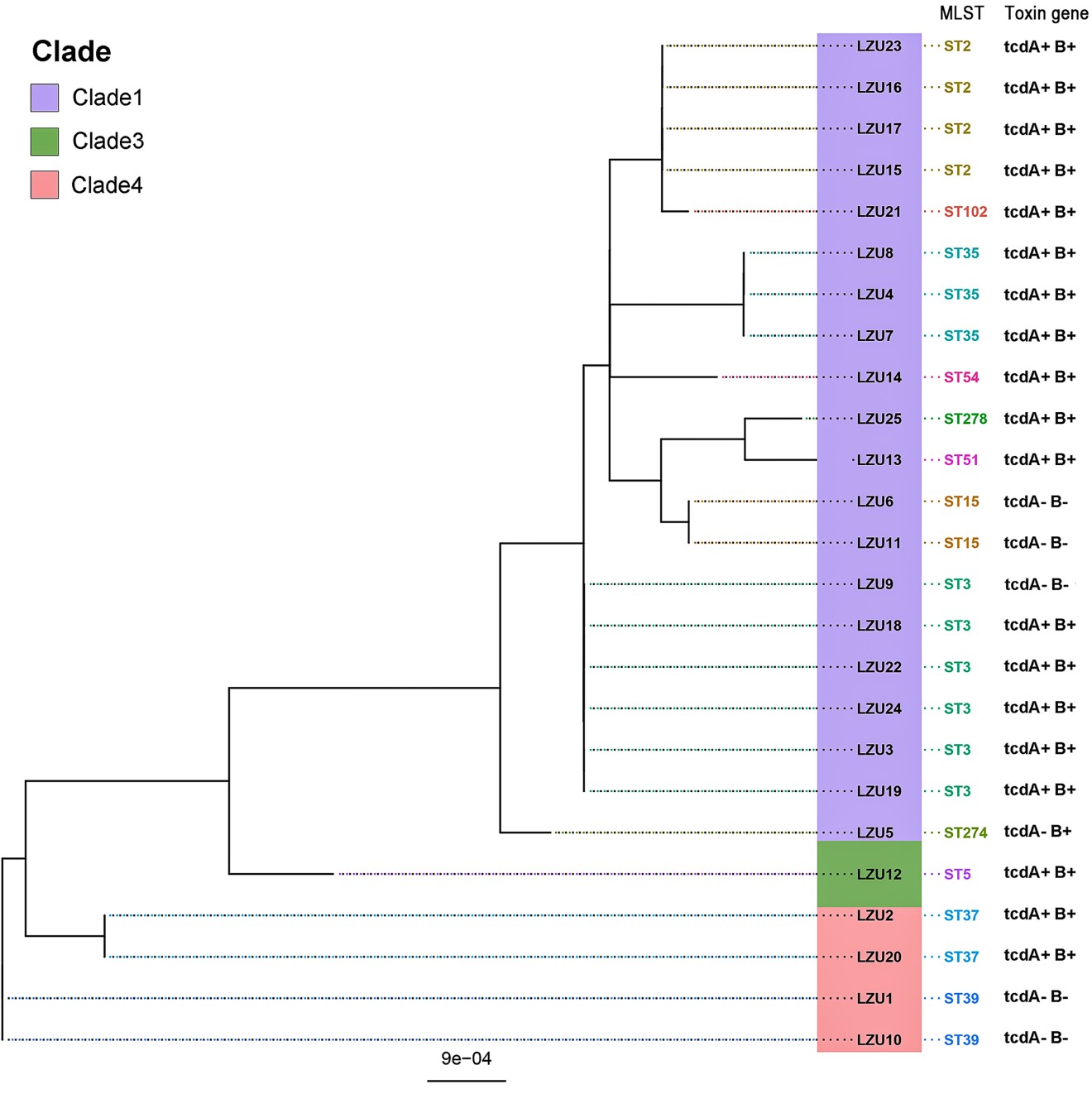

**Figure 1 Phylogenetic tree based on MLST and toxin genes analysis results of 25 *C. difficile* strains.**

two decades, the incidence of CDI has risen due to the widespread use of broad-spectrum antimicrobials, aging populations, and advancements in life-support techniques (*Czepiel et al., 2019*; *Owens et al., 2008*). Studies have reported that the prevalence of *C. difficile*
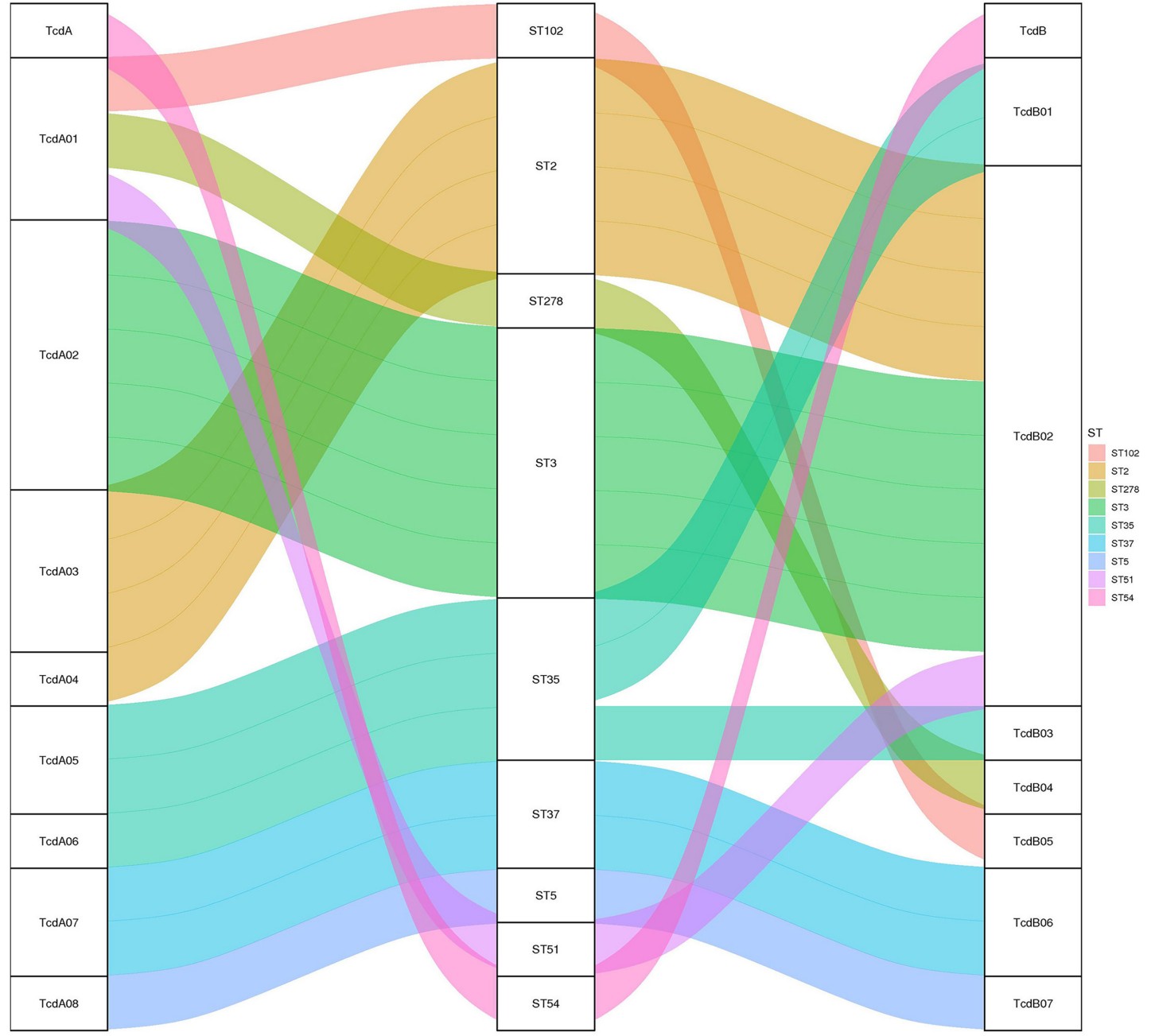

**Figure 2 The relationship between *tcdA*/*tcdB* variants and MLST.** A Sankey diagram showing how *tcdA* and *tcdB* variants are associated with different MLST strains, color-coded for clarity.

among hospitalized patients with diarrhea in Asian countries ranges from 7.1% to 45.1% (*Angulo et al., 2024*). In China, the prevalence of toxin-producing strains of *C. difficile* ranges from 68.2% to 91.9% (*Angulo et al., 2024*). Despite the high prevalence, clinical testing for *C. difficile* remains limited in China due to a lack of awareness among clinicians and challenges associated with anaerobic culture conditions (*Wen et al., 2023*). Consequently, the prevalence of *C. difficile* in various regions is not well-defined, and

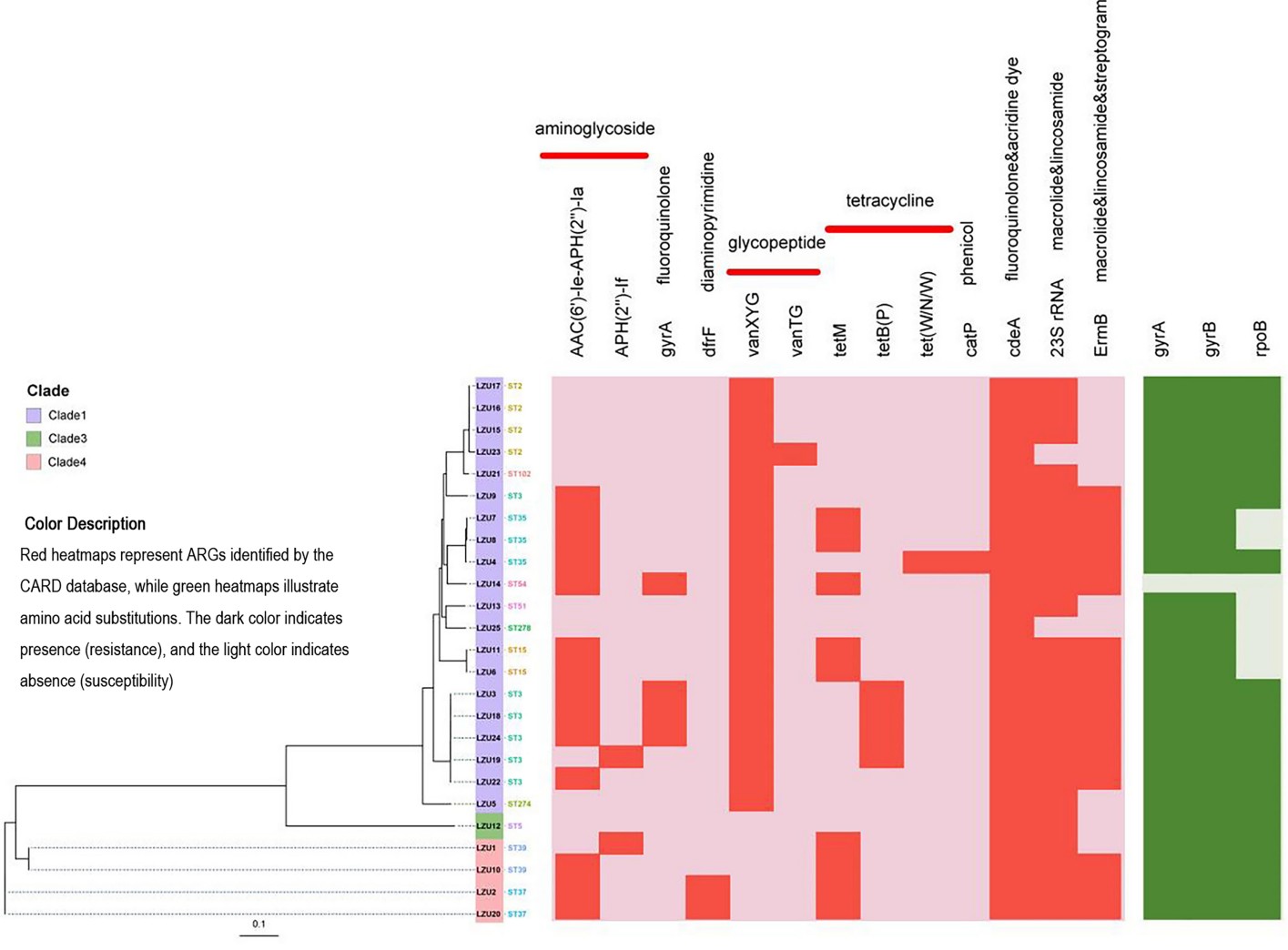

**Figure 3** **Phylogenetic tree constructed by calling SNPs and heat map of the antimicrobial resistance elements.** *C. difficile* strain M68 served as the control.

knowledge about *C. difficile* genes based on WGS is limited (*Wen et al., 2022*). In this study, we prospectively collected fecal samples from patients with diarrhea at a teaching hospital in Gansu, China, to obtain an initial understanding of the epidemiological characteristics of *C. difficile* in this region. Our findings reveal that the detection rate of *C. difficile* in patients with diarrhea was 11.8%, and the positivity rate for *C. difficile* toxin genes was 10.4%. These results are consistent with findings from other regions in China (*Tang et al., 2016*). Notably, most infected patients had been on antibiotics before testing, reinforcing the strong association between antibiotic use and CDI.

In epidemiologic studies of CDI, PCR-ribotyping has been regarded as the reference standard for molecular typing of pathogens (*Wilcox et al., 2012*). However, it lacks the discriminatory power required for detailed analyses of transmission and outbreaks (*Baktash et al., 2022*). Over the past decade, highly virulent strains of RT027 have been

**Table 3 13 drug resistance genes contained in the strains.**

| Antimicrobial resistance genes | Categories of antibiotics | Number of strains containing resistance genes (%) ($n = 25$) |
|---|---|---|
| AAC(6′)-Ie-APH(2″)-Ia | Aminoglycoside | 14 (56.0) |
| APH(2")-If | Aminoglycoside | 2 (8.0) |
| gyrA | Fluoroquinolone | 4 (16.0) |
| dfrF | Diaminopyrimidine | 2 (8.0) |
| vanXYG | Glycopeptide | 20 (80.0) |
| vanTG | Glycopeptide | 1 (4.0) |
| tetM | Tetracycline | 9 (36.0) |
| tetB(P) | Tetracycline | 4 (16.0) |
| tet(W/N/W) | Tetracycline | 1 (4.0) |
| catP | Phenicol | 1 (4.0) |
| cdeA | Fluoroquinolone&acridine dye | 25 (100.0) |
| 23S rRNA | Macrolide & lincosamide | 23 (92.0) |
| ermB | Macrolide & lincosamide & streptogramin | 15 (60.0) |

**Table 4 Significant associations between resistance genes and antibiotic use.**

| Resistance gene | Antibiotic | p-value (Fisher's exact test) | Odds ratio (OR) |
|---|---|---|---|
| ErmB | Penicillins | 0.00012 | 6.5 |
| gyrA | Quinolones | 0.00005 | 7.8 |

**Note:**
p-value: calculated using Fisher's exact test, significant if < 0.000641 (Bonferroni-corrected threshold). Odds ratio (OR): OR > 1 indicates higher odds of gene presence with antibiotic use.

identified in some regions of China, but have not caused large-scale outbreaks (*Cheng et al., 2016*). With the wide application of MLST and WGS, researchers have found that the results of ribotyping are not completely consistent with those of MLST (*Killgore et al., 2008*). In China, prevalent *C. difficile* strains differ from those found in other countries, with MLST types ST54, ST3, and ST37 being the most common, and ST2 predominating in northern China (*Wen et al., 2023*; *Wu et al., 2022*). Our study using WGS identified a diverse range of *C. difficile* STs, with ST3, ST2, and ST35 being the most prevalent. Notably, 80% of the strains belonged to Clade 1, consistent with previous studies from mainland China and highlighting regional prevalence patterns (*Wen et al., 2023*). Importantly, we did not detect the highly virulent strains ST1/RT027 and ST11/RT078 in our sample.

MLST and SNP analysis are complementary methods used to investigate the genetic relationships among bacterial strains. MLST offers a rapid and straightforward approach to identify and compare strains based on allelic variations in housekeeping genes (*Kozyreva et al., 2017*), while SNP analysis provides higher resolution by examining nucleotide-level differences across the entire genome, revealing finer details of genetic relatedness (*Schürch et al., 2018*). In this study, we applied both techniques to assess the genetic diversity of *C. difficile* isolates from Gansu, China. Using the reference genome of

*C. difficile* strain M68 (*Shu et al., 2023*), we conducted SNP analysis on all isolates and constructed a phylogenetic tree to explore their evolutionary relationships. The SNP analysis results were generally consistent with MLST typing outcomes obtained from WGS, with most isolates clustering according to their STs. However, strain LZU9, classified as ST3 by MLST, deviated from this pattern, showing low genetic homology with other ST3 strains in the SNP-based phylogenetic tree. Furthermore, unlike typical ST3 strains that carry both *tcdA* and *tcdB* toxin genes, strain LZU9 exhibited a tcdA−B− toxin profile, underscoring its distinctiveness. This exception demonstrates the advantage of integrating MLST with SNP analysis to detect subtle genetic variations that may influence strain characteristics and epidemiology.

TcdA and TcdB play crucial roles in the pathogenesis of *C. difficile* infections. Previous research indicates that the highly virulent strain ST1/RT027 produces higher levels of TcdA and TcdB toxins in addition to CDT toxins (*Walk et al., 2012*). In this study, PCR detection revealed that the majority of *C. difficile* isolates (80%) were *tcdA+B+*. The inconsistency between the PCR assay and the WGS toxin analysis of the ST274 strain may be attributable to issues with genome assembly or incomplete database annotation, which could lead to either the failure to identify toxin genes or their inaccurate annotation. MLST and virulence genotypes generally matched for the strains, with the exception of ST3. Notably, while ST37 strains typically exhibit a *tcdA−B+* profile in other studies (*Okada et al., 2020*), both ST37 strains isolated in our study were *tcdA+B+*. These discrepancies indicate the genetic diversity of *C. difficile* strain genes. This highlights the importance of using a combination of methods for accurate virulence gene detection. Previous research had suggested that variants of *tcdA* and *tcdB* may affect the biological properties of the toxins, such as enzyme activity, immunogenicity, and receptor affinity (*Mansfield et al., 2020*; *Shen et al., 2020*). Our analysis of *tcdA* and *tcdB* variants demonstrated a strong concordance between toxin variants and MLST types. This finding suggests that *tcdA* and *tcdB* are stable in *C. difficile* and infrequently undergo mutations.

By analyzing the CARD database, we identified 13 genes significantly associated with resistance to multiple antibiotic classes. Notably, we observed that the gene c*deA*, which confers resistance to fluoroquinolone and macrolide antibiotics, was present in all tested strains. Additionally, mutations in the *C. difficile* 23S rRNA gene, which lead to resistance against erythromycin and clindamycin, were prevalent in the majority of the strains. This observation is consistent with previous studies reporting *C. difficile* resistance to ciprofloxacin, erythromycin, and clindamycin (*Huang et al., 2009*). Importantly, all strains within clade 1 carried the vancomycin resistance gene *vanXYG*, a mutated variant of *vanXY* in the *vanG* gene cluster. Despite the presence of *vanXYG*, phenotypic testing is necessary to confirm resistance, which was not performed in this study. The detection of tetracycline resistance genes *tetM* (36%) and *tetB(P)* (16%) in our isolates aligns with findings from Chongqing, China, where *tetB(P)* was identified in tigecycline-resistant *C. difficile* strains (*Dang et al., 2024*), suggesting its potential role in conferring low-level resistance to tetracycline-class antibiotics. Both genes were predominantly linked to

Clade 1 strains (ST3/ST2), indicating clonal expansion as a driver of resistance dissemination. Further analysis revealed that mutations in the *gyrA*, *gyrB*, and *rpoB* genes were also common, potentially contributing to resistance against fluoroquinolones and rifamycins (*Pecavar et al., 2012*). These findings emphasize the potential risk of *C. difficile* developing resistance under antibiotic selection pressure. The significant associations between resistance genes and antibiotic use provide insights into the selective pressures driving *C. difficile* resistance. For instance, the correlations between *ermB* and penicillin use, and *gyrA* and quinolone use are consistent with known resistance mechanisms, indicating that antibiotic exposure may promote the retention of these genes.

Objectively, this study has several limitations. Firstly, the sample size was relatively small and restricted to a single hospital, which may limit the generalizability of the findings to other institutions in China. To address this limitation, future research should involve large-scale, multicenter studies. Secondly, although this study primarily focused on the genetic characteristics of *C. difficile*, these findings should be validated through subsequent phenotypic studies to ensure comprehensive understanding. Thirdly, the small sample size limited the power of statistical analyses to detect correlations between resistance genes and clinical variables, emphasizing the need for larger cohorts in future studies. Finally, clinical outcome data (*e.g.*, severity, recurrence, mortality) were not collected, limiting our ability to evaluate the clinical impact of *C. difficile* infections. Future research will prioritize the inclusion of such data to enhance clinical relevance.

## CONCLUSIONS

This study provides a comprehensive analysis of the epidemiology and genetic characteristics of *C. difficile* isolates from a tertiary hospital in Northwest China. We identified a *C. difficile* isolation rate of 11.8% among 212 patients with diarrhea, with 80% of isolates being toxigenic, predominantly carrying *tcdA+B+* toxin profiles. MLST revealed a diverse array of 12 sequence types, with ST3, ST2, and ST35 being the most prevalent, all belonging primarily to Clade 1, consistent with regional patterns in China. Notably, hypervirulent strains ST1/RT027 and ST11/RT078 were absent, suggesting a distinct regional epidemiology. WGS further elucidated the genetic diversity through SNP analysis and identified 13 antimicrobial resistance genes, including *cdeA* (100%) and *tetM* (36%), highlighting the potential for multidrug resistance under antibiotic selection pressure. These findings underscore the need for enhanced surveillance and targeted infection control strategies to mitigate CDI in hospital settings. Despite limitations such as a single-center design and the absence of antimicrobial susceptibility testing, this study lays a foundation for future multicenter research to validate these findings and explore phenotypic resistance profiles, ultimately informing effective CDI prevention and treatment strategies in Northwest China.

## ACKNOWLEDGEMENTS

We thank all the researchers in this study for their dedication.

### Funding

This work was supported by the Youth Technology Foundation of Gansu Province (No: 21JR11RA073), the Natural Science Foundation of Gansu Province (No: 24JRRA297), and the Youth Fund of the first hospital of Lanzhou University (No: ldyyyn2020-43). The funders had no role in study design, data collection and analysis, decision to publish, or preparation of the manuscript.

### Grant Disclosures

The following grant information was disclosed by the authors:
Youth Technology Foundation of Gansu Province: 21JR11RA073.
Natural Science Foundation of Gansu Province: 24JRRA297.
Youth Fund of the First Hospital of Lanzhou University: ldyyyn2020-43.

### Competing Interests

The authors declare that they have no competing interests.

### Author Contributions

- Yang Li conceived and designed the experiments, performed the experiments, analyzed the data, prepared figures and/or tables, authored or reviewed drafts of the article, and approved the final draft.
- Shujuan An performed the experiments, prepared figures and/or tables, and approved the final draft.
- Hongjia Sun analyzed the data, prepared figures and/or tables, and approved the final draft.
- Meimei Hu analyzed the data, authored or reviewed drafts of the article, and approved the final draft.
- Yanmei Xu performed the experiments, authored or reviewed drafts of the article, and approved the final draft.
- Yaming Xi conceived and designed the experiments, authored or reviewed drafts of the article, and approved the final draft.

### Human Ethics

The following information was supplied relating to ethical approvals (*i.e.*, approving body and any reference numbers):

The Ethics Committee of the First Hospital of Lanzhou University approved the study (Approval number: LDYYLL2022-13).

### Data Availability

Data is available at NCBI: PRJNA1160909.

## Supplemental Information

Supplemental information for this article can be found online at http://dx.doi.org/10.7717/peerj.19877#supplemental-information.

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
