# Peer review of "Epidemiology and genetic characteristics of *Clostridioides difficile* isolates in Northwest China"

_PeerJ, doi:10.7717/peerj.19877_

## Round 0.1 · original submission · Major Revisions

Please address all authors comments.

Reviewer 1 ·

Basic reporting

The present study was conducted to investigate the presence of C. difficile in faecal samples collected from patients hospitalized at the hospital. Furthermore, the genetic characterisation of the C. difficile isolates was performed to detect AMR genes or mutated genes. The epidemiology of the pathogen was investigated by defining the MLST and the genetic diversity, and the relationships between isolates were studied.
- It is evident that English has not yet reached its optimal standard, but it is nonetheless sufficiently clear for the intended purpose to be fulfilled.
- It is evident that the literature references, background, and context are sufficient for the present manuscript.
- The article's structure, figures, and tables are consistent with a conventional format comprising the following sections: introduction, methodology, results, discussion, and conclusion. The raw data were not deposited in any of the genome databases, such as NCBI. It must be submitted to the NCBI database.
- The results of the present manuscript were relevant to the studied hypothesis; however, certain findings remained absent. These included the PCR-ribotyping of the isolates obtained and the antimicrobial susceptibility testing of the important antibiotics, i.e., vancomycin, moxifloxacin, metronidazole, tetracycline, and rifampicin.
Here points that should be addressed:
Line 22: correct. Twenty-five (11.8, 25/212), please include the number of samples to be clearer.
line 23: (9.4, 20/212)
Line 26-27: tcdA, tcdB must be italic.
Line 29: In the abstract, the authors should supply more detailed information on the predominant AMR genes and the antibiotic classes to which they belong.
Line 53: replace “pathogenicity determining region” with “19.6-kb pathogenicity locus”

Experimental design

Some of the methods applied in the current study need more information, as well as some important methods not applied in the methodology section, such as antimicrobial susceptibility testing (AST) and PCR-ribotyping. Here are some points that should be considered:
- The author identified the mutated rpoB gene, which is responsible for encoding rifampicin resistance, in the genome sequences of C. difficile. However, it should be noted that this gene is not always expressed, despite being located in the genome of bacteria. Mutations in rpoB have been observed to influence the correlation with resistance to rifampicin. Consequently, the examination of the phenotype of rifampicin is imperative to establish a correlation between phenotype and genotype.
- The authors also mentioned that virulent ribotypes RT027 and RT078 were not detected in the current isolates (See lines 154-155). It is imperative to ascertain whether PCR-ribotyping was not applied. Please provide clarification in the materials and methods section.
- In relation to the isolation and identification of C. difficile from faecal specimens, further details are required concerning the dilution of the faecal samples. The morphology of colonies on the appropriate media should be addressed, such as white-grey colonies and the horse odour. In addition, the stock cultures of the isolates should be included.
- The AST should be included in the methods section.
Line 118-119: The references to web servers, such as MUMmer and VFDB, should be cited.

Validity of the findings

The findings of the current study can be improved. Here are some points that should be addressed:
- The percentage of the nontoxigenic C. difficile in the faecal samples should be included in the results section
- The percentage of binary toxins CDT compared to toxigenic strains should be included in the results section as well, or if they were detected in all toxigenic strains. Please clarify it.
- To facilitate the analysis of antimicrobial resistance (AMR) genes, it is imperative that the research includes additional information regarding the genes that were detected, as well as the percentage of each. Furthermore, it is essential to include the encoding of the genes in question. In addition, the other genes such as tetM, tetB(P), etc, should be included in the result section as well as discussed in the discussion section.
- I suggest that the mobile genetic elements of the genome sequences of C. difficile strains could be identified and their association with AMR genes.
- SNP analysis was not included in the results section. Please clarify.
- Conclusions could be improved.
Line 144: C. difficile is italic. Please check and correct in all manuscript.
Line 151: For what purpose does the reference to Knight et al. (2015) appear in this section of the text, when the results of STs are the sole subject of discussion?
Line 166: In the subtitle AMR genes analysis, the appropriate figure (Maybe Fig. 3) should be cited here.
Line 171-172: Correct. "The RpoB subunits of RNA polymerase confer mutational resistance to rifampicin".
Line 173: The gyrA and gyrB genes are italic, please correct.
Line 208-217: It is imperative that this paragraph concerning the SNP analysis and the MLST be rewritten. The clarity of this section was found to be inadequate.
Line 253: Cdea replace with cdeA
Line 242: Here, the authors wrote that … “Despite this, these strains remained susceptible to vancomycin”.. In the absence of an application of the AST, it is not possible to determine whether the isolates are susceptible to vancomycin. It is imperative to acknowledge that this is not a complete cluster of vancomycin resistance.
Line 254: replace and with or.

Reviewer 2 ·

Basic reporting

The manuscript is generally written in clear and professional English. The introduction provides a solid contextual background and clearly outlines the relevance of C. difficile infections globally and in China. Literature is generally appropriate and well-referenced. The manuscript follows a standard scientific structure. Figures and tables are appropriately referenced, although Figure 1 should be provided in higher resolution, and labeling in Figure 3 should be improved for better clarity; consider incorporating color descriptions from the figure legend directly into the figure as labels.

Experimental design

The study fits well within the scope of PeerJ and provides a valuable contribution to our understanding of C. difficile epidemiology in China, particularly in a region with limited genomic data. The use of whole-genome sequencing (WGS) and multilocus sequence typing (MLST) is appropriate and methodologically sound. Most methods are described with sufficient detail to allow for replication.
However, some procedural details are underdeveloped (e.g., how SNP thresholds were defined in phylogenetic analysis, how clustering or diversity metrics were calculated). Could the authors provide more detail regarding data normalization/quality control in bioinformatics analyses?
Furthermore, a dedicated subsection describing the clinical data would be beneficial. This should include clear definitions for variables such as 'antibiotic use prior to diarrhea,' specifying the timing and duration of antibiotic exposure, as well as clarification on the classification of comorbid conditions

Validity of the findings

The study's findings are valid and support the conclusion that a genetically diverse array of C. difficile genotypes circulates within this hospital, with a predominance of non-hypervirulent clade 1 strains. The absence of ST1/RT027 and ST11/RT078 is a notable regional finding and aligns with previous reports from other parts of China. The interpretation of the data is generally sound. The inclusion of virulence gene variants and antimicrobial resistance gene profiles adds value and novelty to the work. The authors acknowledge the study’s limitations, including its single-center setting and relatively small sample size.

Additional comments

- In Table 1, please clarify "antibiotic use prior to Diarrhea" by listing the different antibiotics used. If possible, include a comparison between the total patient population and C. difficile-positive patients, with statistical testing to assess significant differences across clinical variables. What about the clinical setting (e.g., ICU vs. general wards)? Was this data collected, and could it be added to Table 1/manuscript?

- Consider performing additional statistical analyses to correlate resistance genes with clinical data.

- Inclusion of clinical outcomes (e.g., severity, recurrence, mortality) would enhance the clinical relevance of the findings.

---

## Round 0.2 · Minor Revisions

Please address the few new comments of the reviewer.

Reviewer 1 ·

Basic reporting

Following a thorough review of the current manuscript, gratitude is extended to all authors for their contributions to the improvement of the manuscript, including the literature reviews, background, article structure, figures, tables, and so further. It is evident that all previous common concerns have been addressed, as well as any suggestions that were put forward.

Experimental design

All previous comments and suggestions were intended to enhance the current manuscript.

Validity of the findings

I have one comment:
Line 208: Why do you cite the figure 3, in the subtitle, it must be cited in the text. Please clarify.
Line 209: ….” we identified a total of 13 antibiotic resistance genes (ARGs) associated with resistance to various classes of antibiotics including aminoglycosides, fluoroquinolones, diaminopyrimidines, glycopeptides, tetracyclines, macrolides, lincosamides, and streptomycin”.
Could you please include more details and percentage of detected resistance genes such as tetM, tetB(P); ermB…etc, which the dominant ARGs?

Additional comments

Line 288: Correct, ST274, no space.
Line 293: C. difficile is italic.
Line 319: replace “ErmB” with “ermB”

Line 341: single nucleotide polymorphism (SNP), here you should use the abbreviation if you written the full name before (SNP).

---

## Round 0.3 · Minor Revisions

Please address the minor comments of the remaining reviewer-

Reviewer 1 ·

Basic reporting

I have no further comments to add to the current manuscript. The authors have addressed all comments and suggestions.

Experimental design

I have no further comments to add to the current manuscript. The authors have addressed all comments and suggestions.

Validity of the findings

I have no further comments to add to the current manuscript. The authors have addressed all comments and suggestions.

Additional comments

I have no further comments to add to the current manuscript. The authors have addressed all comments and suggestions.

Reviewer 2 ·

Basic reporting

The resubmitted manuscript remains well written in clear and professional English. As noted in my initial review, the introduction continues to provide a solid contextual background and effectively outlines the global and national relevance of C. difficile infections. The literature cited is appropriate and well-referenced, and the manuscript maintains a logical and standard scientific structure. Figures and tables are appropriately referenced. I would like to reiterate my previous suggestion that Figure 1 should be provided in higher resolution.

Experimental design

The study remains well aligned with the scope of PeerJ and continues to offer a valuable contribution to the understanding of C. difficile epidemiology in China, particularly in a region where genomic data have been limited. The application of whole-genome sequencing (WGS) and multilocus sequence typing (MLST) remains appropriate and methodologically sound. Most methods are described in sufficient detail to allow for replication. The dedicated subsection describing the clinical data in the results section 'In this study, we collected comprehensive clinical data to provide context for the epidemiological and genetic characteristics of C. difficile isolates. Antibiotic Use Prior Diarrhea: Antibiotic exposure was defined as the administration of any systemic antibiotic within 30 days preceding the onset of diarrhea. Comorbidities were classified according to the International Classification of Diseases, 10th Revision (ICD-10).' should be added to the methods section.

Validity of the findings

The study’s findings are valid and continue to support the conclusion that a genetically diverse array of C. difficile genotypes circulates within this hospital, with a predominance of non-hypervirulent clade 1 strains. The absence of ST1/RT027 and ST11/RT078 remains a notable regional observation and is consistent with previous reports from other parts of China. The interpretation of the data is generally sound. The inclusion of virulence gene variants and antimicrobial resistance gene profiles continues to add value and novelty to the work. The authors appropriately acknowledge the study’s limitations, including its single-center setting and relatively small sample size.

---

## Round 0.4 · accepted · Accept

Thanks for addressing all reviewers' comments!

Reviewer 1 ·

Basic reporting

It is clear that sufficient attention was paid to the background and introduction, the context, the structure, the figures and the tables of the current manuscript.

Experimental design

The present study demonstrated an adequate response to the experimental design and research questions.

Validity of the findings

The present study concluded with a clear articulation of its findings and the provision of the relevant data. The conclusions of the present study were found to be associated with the research questions and the findings.

Additional comments

No further comments are offered.